

# Genetic diversity analysis and molecular characteristics of wild centipedegrass using sequence-related amplified polymorphism (SRAP) markers

Xiaoyun Wang[1], Wenlong Gou[2], Ting Wang[1], Yanli Xiong[1], Yi Xiong[1], Qingqing Yu[1], Zhixiao Dong[1], Xiao Ma[1], Nanqing Liu[3] and Junming Zhao[1]

[1] College of Grassland Science and Technology, Sichuan Agricultural University, Chengdu, China
[2] Sichuan Academy of Grassland Sciences, Chengdu, China
[3] Jiangsu Vocational College Agriculture and Forestry, Gourong, China

Corresponding authors
Nanqing Liu, Liunanqing@jsafc.edu.cn
Junming Zhao, junmingzhao163@163.com

## ABSTRACT

Centipedegrass (*Eremochloa ophiuroides* (Munro) Hack.) is commonly used as a low-maintenance warm-season turfgrass owing to its excellent adaptation to various soil types. A better understanding of the genetic diversity pattern of centipedegrass is essential for the efficient development and utilization of accessions. This study used fifty-five pairs of primers to detect the genetic variation and genetic structure of twenty-three wild centipedegrass accessions by Sequence-related amplified polymorphism (SRAP) markers. A total of 919 reliable bands were amplified, among which 606 (65.80%) were polymorphic and 160 (2.91%) were the monomorphic loci. The average polymorphic information content (PIC) value was 0.228. The unweighted pair group method with arithmetic mean (UPGMA) clustering analysis grouped the twenty-three accessions into two clusters. Meanwhile, the structure analysis showed that the tested accessions possessed two main genetic memberships ($K = 2$). The Mantel test significantly correlated the genetic and geographic distance matrices ($r = 0.3854$, $p = 0.000140$). Furthermore, geographical groups showed moderate genetic differentiation, and the highest intragroup genetic diversity was found in the Sichuan group ($He = 0.201$). Overall, the present research findings could promote the protection and collection of centipedegrass and provide comprehensive information to develop novel breeding strategies.

## INTRODUCTION

Centipedegrass [*Eremochloa ophiuroides* (Munro) Hack.] is a perennial warm-season diploid grass species (2n = 2x = 18) that belongs to the genus *Eremochloa* in the family Poaceae. Centipedegrass originated in southwest China, and the wild population is reportedly mainly distributed in the southern Yangtze River region of China (*Hanna & Burton, 1978*; *He et al., 2022*). Centipedegrass has the characteristics of beautiful leaf color, low plant height, drought and barren resistance, high coverage, strong disease resistance, and strong adaptability in acidic and slightly alkaline soils (*Liu et al., 2023*; *Li et*

*al., 2022*). It is an ideal broad-leaved grass species, suitable for the construction of sports and leisure lawns, and low maintenance and management requirements (*Cai et al., 2022*; *Li et al., 2020*). It grows slowly, has short stolons and many leaves, and can form a denser lawn. It is one of the early excellent lawn grasses in southern China (*Xu et al., 2023*; *Cai et al., 2022*). In addition, it can consolidate soil, protect slopes and embankments, and prevent soil and water loss, which plays an important role in slope vegetation restoration (*Islam & Hirata, 2010*; *Liu et al., 2008*). Therefore, centipedegrass is a pioneer plant for slope ecological restoration. Although centipedegrass is widely distributed in China, with diverse populations and great potential for development, there are few varieties adapted to specific regional climates, which poses a real challenge to the demand for new centipedegrass varieties with long green periods, high overwintering rates, and adaptation to climate change in non-local environments.

Evaluation of genetic diversity in germplasm resources can provide useful information for plant breeding programs (*Gawali, Bhoite & Pardeshi, 2006*; *AlKhayri et al., 2023*; *AlKhayri et al., 2022*). Analysis of the genetic variation in various markers such as morphology, agronomic traits, and DNA molecular markers showed significant differences between different accessions and populations (*Xuan, Gao & Liu, 2005*; *Zhao, Bai & Liang, 2011*; *Milla-Lewis et al., 2012*). Compared with other biochemical markers, DNA molecular markers have superior characteristics, such as higher polymorphism, more accurate experimental results, and independence from environmental conditions and developmental stages (*Massa et al., 2001*). Furthermore, they represent a robust and quick approach to detecting the genetic variability of germplasm. Over the years, several molecular markers like amplified fragment length polymorphism (AFLP), random amplified polymorphic DNA (RAPD), and inter-simple sequence repeat (ISSR) have been used to elucidate the genetic diversity of centipedegrass accessions (*Xuan, Gao & Liu, 2005*; *Zhao, Bai & Liang, 2011*; *Milla-Lewis et al., 2012*; *Massa et al., 2001*). Sequence-related amplified polymorphism (SRAP) is a new PCR-based approach whereby two sets of primers are designed based on the G and C contents in gene exons to amplify the open reading frame (*Li & Quiros, 2001*; *Robarts & Wolfe, 2014*). Compared with other common dominant markers, it is easier to operate, low-cost, and more functional. Therefore, in recent years, SRAP molecular marker technology has been widely used for the study of genetic diversity in a large number of grass species, such as *Russian Alfalfa* (*Shamustakimova, Mavlyutov & Klimenko, 2021*), *Buchloe dactyloides* (*Wu et al., 2019*), *Dactylis glomerata* (*Zeng et al., 2008*).

Few studies have hitherto used SRAP to explore the genetic diversity of centipedegrass accessions. This study combined SRAP molecular markers with the seven morphological indexes to reveal the genetic and morphological diversity of twenty-three centipedegrass accessions. This study aimed to reveal the population genetic structure of these materials at the molecular level. In addition, morphological diversity analysis was conducted to obtain more comprehensive information, which is of great significance for preserving valuable genetic resources, selecting high-quality germplasm resources, and developing new varieties.

## MATERIAL AND METHODS

### Plant samples and DNA extraction

A total of twenty-three wild centipedegrass accessions were collected from Sichuan province ($n = 9$), Chongqing municipality ($n = 6$), abroad ($n = 1$), and other parts of China ($n = 7$) (Table S1, Fig. S1). In early May 2016, seven morphological traits were measured and scored in the experimental field of Hanchang town, Chengdu city in China (30°35′24″N, 103°31′48″E), which were erect branch leaf length(EBLL), erect branch leaf width (EBLW), stolon leaf length (SLL), stolon leaf width (SLW), stolon internode length (SIL), stolon internode diameter (SLD), and grass height (GLH) (Table S2). We divided the 23 accessions into three groups according to their geographical origins: Sichuan (nine accessions), Chongqing (six accessions), and Other areas (eight accessions). Dispersed geographical groups with few individuals were classified into the same group.

The above seven morphological traits are measured as follows: For EBLL and EBLW, five mature leaves were randomly selected from upright branches and the length and width (at the widest point) of the leaves were determined using a vernier caliper. For SLL and SLW, five mature leaves were randomly selected from the stolons and the length and width(at the widest point) of the leaves were determined using vernier calipers. For GLH, randomly measure the natural height of the grass layer and repeat the measurement 5 times. For SIL and SID, random five healthy stolons were selected to determine the internode length and diameter in the middle of stolons.

Genomic DNA was extracted using a Plant Genomic DNA Extraction Kit(DP305, Beijing Tiangen). The concentration of DNA was detected by ultramicro spectrophotometer. Completely tested DNA samples were diluted to 10 ng/μL with sterile ddH$_2$O and stored at −20 °C for PCR amplification.

### SRAP analysis

A total of 215 pairs of SRAP primers were randomly combined to screen polymorphic primers for twenty-three wild centipedegrass accessions. SRAP amplification system: 15 μL SRAP reaction system: DNA template 3 μL (10 ng μL$^{-1}$), MIX 7.5 μL (dNTP 240 μmol L$^{-1}$, Taq enzyme 1.0 U μL$^{-1}$, Mg$^{2+}$ 2.5 mmol L$^{-1}$), upstream and downstream primer 0.3 μL (10 μmol L$^{-1}$) each, ddH$_2$O 3.6 μL, and Taq enzyme 0.3 μL. The SRAP-PCR reaction was performed as follows: predenaturation at 94 °C for 5 min, 5 cycles of denaturation at 94 °C for 1 min, annealing at 35 °C for 1 min, stretching at 72 °C for 1 min, 35 cycles of denaturation at 94 °C for 1 min, annealing at 50 °C for 1 min, 72 °C for 1 min, final extension at 72 °C for 10 min and storage at 4 °C. The PCR products were separated by 6% modified polyacrylamide gel and detected by silver staining. Gel clear photographs were used for the following analysis.

### Data analysis

The polymorphic bands were statistically analyzed according to the electrophoresis results. The presence and absence of stripes were recorded as 1 and 0, respectively. Finally, a (0, 1) matrix was generated for statistical software analysis. The number of polymorphic bands (NPB), percentage of polymorphic bands (PPB), marker index (MI), and resolution (RP)

were calculated to evaluate the ability of SRAP primers to identify marker differences. PIC was used to evaluate the value of markers for detecting population polymorphism. PIC was calculated by the following formula:

$$PIC = 1 - \sum P_i^2.$$

Where Pi is the frequency for the i th microsatellite allele (*Riek et al., 2001*). The GenAlex 6.51 procedure (*Peakall, 2012*) was used to estimate the effective number of alleles (Ne), Shannon information index (I), and pairwise population PhiPT values (Fst) among the geographical groups. At the same time, principal coordinates analysis(PCoA) was used to analyze the information quality of specific SRAP primers. In addition, NTSYS-pc software was used for cluster analysis of the unweighted pair group method with arithmetic mean (UPGMA), and a tree diagram was generated. The relationship between morphological indexes, climatic data, and genetic similarity coefficients of all germplasms was determined by the Mantel test (*Zeller Katherine et al., 2016*). Otherwise, we further evaluated the genetic structure of the population of twenty-three germplasm resources using the STRUCTURE 2.3.3 software (*Pritchard, Stephens & Donnelly, 2000*), with population K set to 1–10. The number of iterations for the burn-in and post-burn periods was set to $10^4$ and $10^5$ for the Markov chain Monte Carlo simulations. Then the online program was used to determine the optimal K value (*Dent & Bridgett, 2012*).

## RESULTS

### Polymorphism analysis

Fifty-five pairs of qualified primers were screened from 215 pairs of primers, and the polymorphism of 23 wild centipedegrass accessions germplasm resources was evaluated. The results showed that the number of reliable bands amplified by each primer pair was seven (M14E07)–23(M01E07), and a total of 919 reliable bands were amplified. The polymorphic bands per primer pair ranged from 16.67% (M07E07) to 90% (M01E20 and M17E10), with an average of 65.8%. The polymorphism and recognition ability of primers were evaluated by PIC, MI and RP. The average PIC value was 0.228, and the PIC value of primer M12E19 was the highest (0.312). The average MI and RP values were 1.85 and 5.40, respectively, indicating the high utility of the primers (Table 1, Fig. S2).

### Clustering, PCoA, and population structure analysis

Based on the (0, 1) matrix, UPGMA analysis showed that all accessions could be divided into two clusters (Fig. S3). Cluster I was mainly from Chongqing and other areas, and cluster II was mainly from Sichuan. Through principal coordinates analysis, another clustering of twenty-three wild centipedegrass accessions was performed to generate a scatter plot (Fig. 1). The results showed that PCoA divided twenty-three accessions into two clusters. The molecular variation explained by principal coordinate 1 was 14.31%, which was roughly the same as the result of UPGMA tree (the specific PCoA values are detailed in Table S4). A tree map was constructed based on morphological trait data, and all accessions could be divided into two groups at an average distance of 30.399, indicating that they could be grouped independently regardless of geographic distribution (Fig. S4).
**Table 1  Polymorphism of SRAP markers in centipedegrass accessions.**

| Primer Pairs | TNB | NPB | MB | Polymorphism | PIC | RP | MI |
|---|---|---|---|---|---|---|---|
| M01E03 | 14 | 10 | 1 | 71.43 | 0.248 | 4.78 | 1.77 |
| M20E05 | 22 | 14 | 4 | 63.64 | 0.213 | 6.43 | 1.90 |
| M01E14 | 15 | 11 | 2 | 73.33 | 0.199 | 3.83 | 1.61 |
| M01E20 | 10 | 9 | 0 | 90.00 | 0.263 | 3.48 | 2.13 |
| M01E07 | 23 | 16 | 2 | 69.57 | 0.239 | 7.91 | 2.66 |
| M01E09 | 17 | 13 | 1 | 76.47 | 0.281 | 6.87 | 2.79 |
| M06E03 | 18 | 10 | 3 | 55.56 | 0.196 | 4.70 | 1.09 |
| M06E06 | 17 | 7 | 7 | 41.18 | 0.121 | 2.78 | 0.35 |
| M06E07 | 14 | 11 | 2 | 78.57 | 0.224 | 4.43 | 1.93 |
| M06E09 | 21 | 11 | 6 | 52.38 | 0.170 | 4.43 | 0.98 |
| M06E12 | 13 | 9 | 3 | 69.23 | 0.242 | 4.70 | 1.51 |
| M06E19 | 14 | 6 | 4 | 42.86 | 0.164 | 3.04 | 0.42 |
| M06E04 | 19 | 10 | 4 | 52.63 | 0.215 | 6.00 | 1.13 |
| M07E04 | 20 | 10 | 6 | 50.00 | 0.185 | 5.39 | 0.93 |
| M07E09 | 13 | 6 | 6 | 46.15 | 0.136 | 2.35 | 0.38 |
| M07E07 | 12 | 2 | 6 | 16.67 | 0.114 | 2.09 | 0.04 |
| M09E03 | 17 | 8 | 5 | 47.06 | 0.196 | 4.61 | 0.74 |
| M09E10 | 19 | 10 | 4 | 52.63 | 0.208 | 5.65 | 1.10 |
| M09E14 | 17 | 13 | 3 | 76.47 | 0.229 | 5.22 | 2.28 |
| M09E20 | 19 | 15 | 2 | 78.95 | 0.264 | 6.96 | 3.13 |
| M10E03 | 17 | 11 | 3 | 64.71 | 0.238 | 6.43 | 1.70 |
| M10E06 | 15 | 9 | 5 | 60.00 | 0.186 | 3.48 | 1.00 |
| M10E08 | 19 | 15 | 1 | 78.95 | 0.242 | 6.00 | 2.87 |
| M10E15 | 14 | 9 | 3 | 64.29 | 0.227 | 4.43 | 1.32 |
| M11E01 | 23 | 21 | 3 | 80.77 | 0.270 | 9.91 | 4.58 |
| M11E06 | 20 | 17 | 3 | 85.00 | 0.258 | 6.96 | 3.73 |
| M11E09 | 22 | 17 | 3 | 77.27 | 0.285 | 9.39 | 3.74 |
| M11E10 | 17 | 9 | 5 | 52.94 | 0.196 | 4.61 | 0.93 |
| M11E14 | 17 | 11 | 2 | 64.71 | 0.223 | 5.22 | 1.59 |
| M12E02 | 16 | 12 | 1 | 75.00 | 0.272 | 6.09 | 2.45 |
| M12E06 | 15 | 12 | 2 | 80.00 | 0.270 | 5.83 | 2.59 |
| M12E08 | 18 | 11 | 3 | 61.11 | 0.213 | 5.48 | 1.43 |
| M12E09 | 14 | 9 | 2 | 64.29 | 0.263 | 5.30 | 1.52 |
| M12E03 | 18 | 15 | 0 | 83.33 | 0.252 | 6.09 | 3.15 |
| M12E19 | 19 | 15 | 0 | 78.95 | 0.312 | 9.13 | 3.69 |
| M14E14 | 12 | 8 | 2 | 66.67 | 0.255 | 4.26 | 1.36 |
| M14E04 | 13 | 10 | 3 | 76.92 | 0.270 | 5.39 | 2.08 |
| M14E07 | 7 | 3 | 2 | 42.85 | 0.188 | 2.00 | 0.24 |
| M15E15 | 18 | 13 | 1 | 72.22 | 0.270 | 7.13 | 2.54 |
| M15E03 | 17 | 13 | 1 | 76.47 | 0.278 | 6.78 | 2.93 |
| M15E08 | 19 | 13 | 4 | 68.42 | 0.274 | 7.91 | 2.44 |

*(continued on next page)*

**Table 1** (*continued*)

| Primer Pairs | TNB | NPB | MB | Polymorphism | PIC | RP | MI |
|---|---|---|---|---|---|---|---|
| M16E03 | 18 | 10 | 5 | 55.56 | 0.190 | 4.43 | 1.05 |
| M16E08 | 22 | 15 | 5 | 68.18 | 0.233 | 6.78 | 2.38 |
| M16E10 | 14 | 8 | 4 | 57.14 | 0.224 | 4.61 | 1.02 |
| M17E06 | 22 | 11 | 4 | 50.00 | 0.212 | 6.87 | 1.17 |
| M17E08 | 13 | 11 | 1 | 84.62 | 0.264 | 4.78 | 2.46 |

| Primer Pairs | TNB | NPB | MD | PPB | PIC | RP | MI |
|---|---|---|---|---|---|---|---|
| M17E09 | 14 | 12 | 1 | 85.71 | 0.299 | 6.00 | 3.07 |
| M17E10 | 10 | 9 | 1 | 90.00 | 0.298 | 4.09 | 2.41 |
| M17E12 | 13 | 6 | 3 | 46.15 | 0.149 | 2.61 | 0.41 |
| M17E14 | 17 | 12 | 1 | 70.59 | 0.264 | 6.52 | 2.24 |
| M17E04 | 13 | 8 | 3 | 61.54 | 0.202 | 3.57 | 0.99 |
| M18E06 | 19 | 13 | 3 | 68.42 | 0.257 | 7.22 | 2.28 |
| M18E10 | 22 | 17 | 1 | 77.27 | 0.240 | 6.96 | 3.15 |
| M19E12 | 16 | 8 | 3 | 50.00 | 0.188 | 4.00 | 0.75 |
| M20E06 | 22 | 12 | 5 | 54.55 | 0.168 | 5.04 | 1.10 |
| sun | 919 | 606 | 160 | | | | |
| mean | 16.7 | 11.01 | 2.91 | 65.80 | 0.228 | 5.4 | 1.85 |

**Notes.**

TNB, total of bands; NPB, the number of polymorphic bands; MB, monomorphic bands; Polymorphism(%), percentage of polymorphic bands; PIC, polymorphic information content; MI, marker index; RP, resolving power.

The population structure of twenty-three wild centipedegrass germplasms was analyzed by the Bayesian method. When the Evanno method was performed, the optimal ΔK was 2 (Fig. S5). Accordingly, the optimal number of subpopulations in this study was two (namely, two genetic members) (Table S4 and Fig. 1). Assuming that the accessions with a Q value more than 0.8 were "pure" (*Forsberg et al., 2014*), 69.56% of the germplasm was attributed to the pure subgroup. The proportion of mixed germplasm resources in Sichuan was the largest (Table S5).

## Genetic structure of the inferred geographic groups and mantel analysis

The twenty-three wild centipedegrass accessions could be divided into three categories: Sichuan, Chongqing, and other areas. It was found that Sichuan had the highest genetic diversity (He = 0.201, I = 0.312) (Table 2). The analysis of molecular variance (AMOVA) showed that the Fst among geo-groups was 0.115, indicating a moderate degree of genetic differentiation among geo-groups (Table 3). The genetic distance between the three geographic groups was evaluated. We found that the differentiation between Chongqing and Sichuan was the lowest (Fst = 0.051) (Table 4).

Mental analysis showed no correlation between the genetic and morphological distance matrices ($r = -0.0003$, $p = 0.5093$) (Fig. 2, Table S7). When the genetic distance matrix correlated with climate factors, BIO14 (precipitation in the driest month) ($r = 0.2513$, $p = 0.0063$), BIO15 (precipitation seasonality) ($r = 0.2623$, $p = 0.0434$), and BIO17 (precipitation in the arid region) ($r = 0.2354$, $p = 0.0141$) were highly correlated with

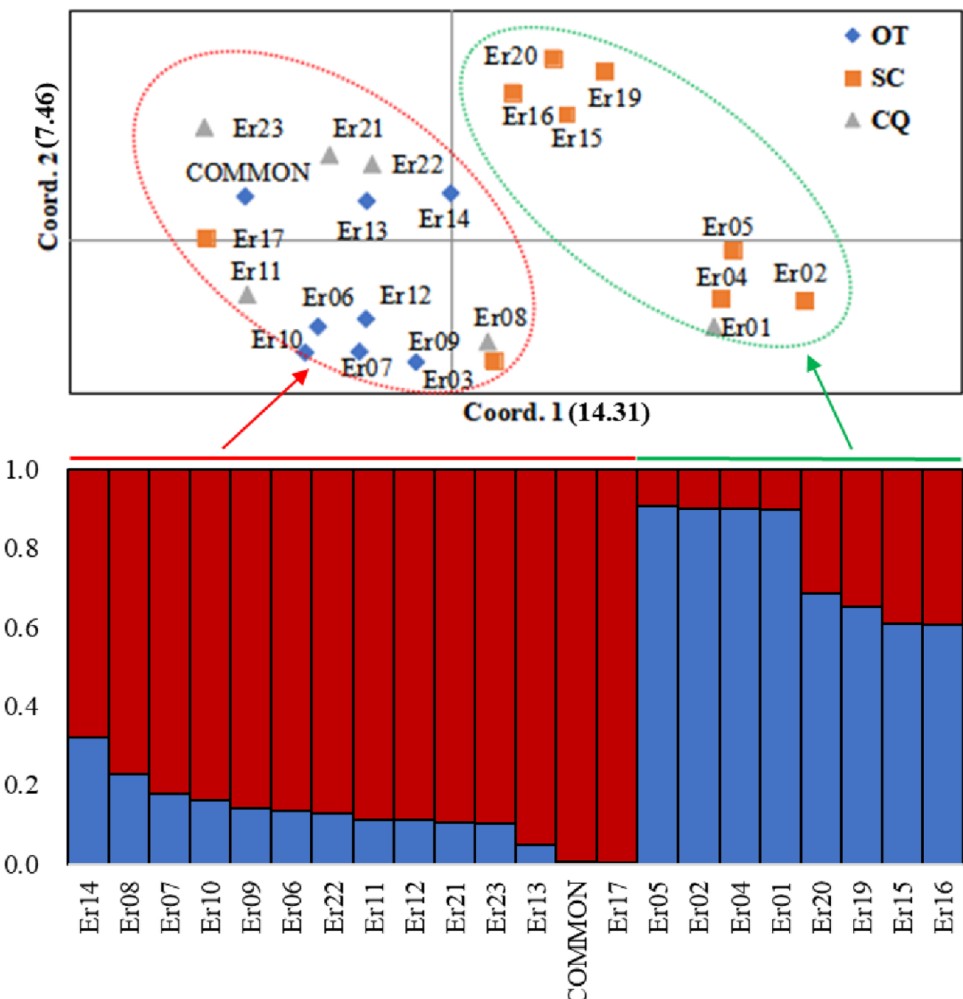

**Figure 1** **PCoA and population structure ($K = 2$) of 23 wild centipedegrass accessions based on SRAP-markers.** SC, from Sichuan province; CQ, from Chongqing municipality; OT, other accessions except Sichuan and Chongqing.

**Table 2** **Genetic diversity assessment for three geographical groups of centipedegrass accessions.**

|        | N     | Na    | Ne    | I     | He    | uHe   |
|--------|-------|-------|-------|-------|-------|-------|
| OT     | 8.000 | 1.145 | 1.220 | 0.206 | 0.134 | 0.143 |
| SC     | 9.000 | 1.491 | 1.326 | 0.312 | 0.201 | 0.213 |
| CQ     | 6.000 | 1.255 | 1.260 | 0.244 | 0.158 | 0.173 |
| Mean   | 7.667 | 1.297 | 1.269 | 0.254 | 0.165 | 0.176 |

Notes.
   N, accessions number; Na, the allele number; Ne, effective number of alleles; I, Shannon information index; He, Expected heterozygosity; uHe, Unbiased expected heterozygosity; SC, from Sichuan province; CQ, from Chongqing municipality; OT, other accessions except Sichuan and Chongqing.

**Table 3  Analysis of molecular variance (AMOVA) based on SRAP markers for geographical groups of centipedegrass accessions.**

| Source | df | SS | MS | Est. Var. | Fst | PMV% |
|---|---|---|---|---|---|---|
| Among geo-groups | 2 | 23.450 | 11.725 | 0.770 | 0.115 | 12% |
| Within geo-groups | 20 | 118.028 | 5.901 | 5.901 | | 88% |
| Total | 22 | 141.478 | | 6.671 | | 100% |

Notes.

df, degree of freedom; SS, square deviation; MS, mean square deviation; Est.Var, exist variance; Fst, coefficient of genetic differentiation; PMV, percentages of molecular variance.

**Table 4  Pairwise population PhiPT values among three geographical groups of centipedegrass accessions.**

| | OT | CQ | SC |
|---|---|---|---|
| OT | 0.000 | | |
| CQ | 0.092 | 0.000 | |
| SC | 0.180 | 0.051 | 0.000 |

Notes.

SC, from Sichuan province; CQ, from Chongqing municipality; OT, other accessions except Sichuan and Chongqing.

genetic distance (Fig. 2, Table S6). At the same time, a correlation was observed between geographical distance and the genetic matrix ($r = 0.385352$, $p = 0.000140$).

## DISCUSSION

Compared with other molecular markers, SRAP has the advantages of simplicity, high efficiency, high yield, and good repeatability (*Budak et al., 2004*; *Gao et al., 2020*; *Li et al., 2019*). The present study, Fifty-five SRAP markers were used to evaluate the genetic diversity of twenty-three wild centipedegrass accessions. Using Fifty-five SRAP markers, 919 scorable fragments were obtained with an average of 16.7 fragments per marker, higher than reported in a study by *Zheng et al. (2017)* in 80 bermudagrass accessions (13.08 fragments per marker) but lower than detected by *Yuan et al. (2018)* in 73 Kentucky bluegrass accessions (18.6 fragments per marker). This finding indicates that the primers screened in this study have very good practicability in studying genetic diversity. Among the 919 fragments, 606 polymorphic bands were found, higher than in an RAPD study (42.41%) by *Xuan, Gao & Liu (2005)*. This finding indicated that centipedegrass germplasm has high genetic diversity.

It is widely acknowledged that the primer efficiency index represents the overall utility of a specific primer during the identification of many accessions; higher values are associated with higher efficiency and amount of information on primers. In this study, the average MI (1.85) and RP (5.40) values of primers were higher than the SRAP and EST-SSR markers in prairie grass (MI = 1.348, RP = 1.897 and MI = 0.67, RP = 1.14) (*Yi et al., 2021*; *Sun et al., 2021*). Besides, the primer pairs M11E01 (MI = 4.58, RP = 9.91) and M11E09 (MI = 3.74, RP = 9.39) had the highest MI and RP values, indicating that these primers had high genetic identification values for centipedegrass germplasm. Besides, the average PIC value of SRAP markers was 0.228, higher than that of *Moonsap et al. (2019)* (PIC = 0.20), indicating the high utility of selected primers.

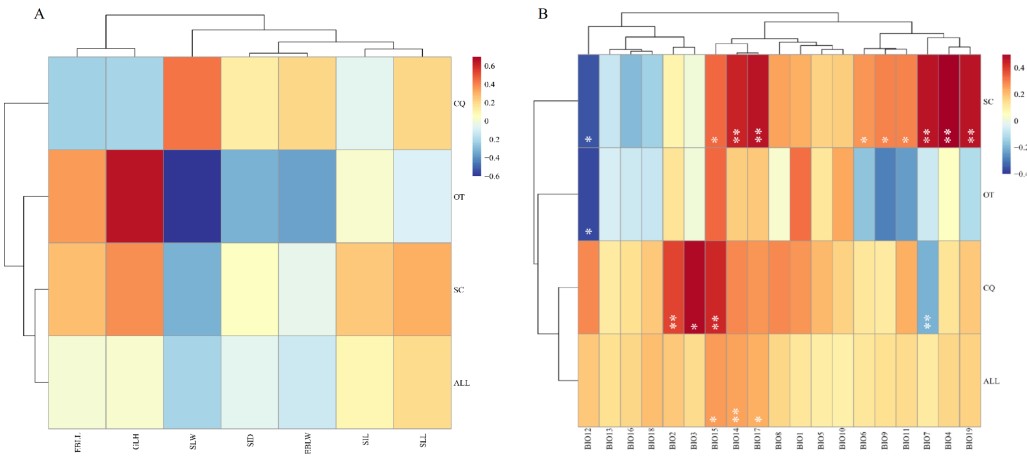

**Figure 2 Heatmap of correlation between genetic distance and morphological traits or genetic distance and climatic factors.** (A) Heatmap of correlation between genetic distance and morphological traits. EBLL, Erect branch leaf length; EBLW, Erect branch leaf; SLL, Stolon leaf length; SLW, Stolon leaf width; SIL, Stolon internode length; SID, Stolon internode diameter. (B) Heatmap of correlation between genetic distance and climatic factors. BIO1, Annual mean temperature; BIO2, Mean diurnal range; BIO3, Isothermality; BIO4, temperature seasonality; BIO5, Max temperature of warmest month; BIO6, Min temperature of coldest month; BIO7, Temperature annual range; BIO8, Mean temperature of wettest quarter; BIO9, Mean temperature of driest quarter; BIO10, Mean Temperature of Warmest Quarter; BIO11, Mean temperature of coldest quarter; BIO12, Annual Precipitation; BIO13, Precipitation of wettest month; BIO14, Precipitation of driest month; BIO15, Precipitation seasonality; BIO16, Precipitation of wettest quarter; BIO17, Precipitation of driest quarter; BIO18, Precipitation of Warmest Quarter; BIO19, precipitation of coldest quarter. SC, from Sichuan province, CQ, from Chongqing municipality, OT, other accessions except Sichuan and Chongqing. Two asterisks (**) indicate very significant difference; an asterisk (*) indicates signficant difference.

In this study, the overall genetic diversity of centipedegrass was higher (He = 0.165) than the average value (0.104) of *Hemarthria compressa* (*Huang et al., 2012*), since the accessions collected in the latter study were concentrated in Yunnan-Guizhou-Sichuan region, the geographical distribution range is limited. *Xuan, Gao & Liu (2005)* used RAPD to study the genetic diversity of centipedegrass and reported an average value lower than in the present study (0.04 *vs.* 0.165), which may be attributed to the collection of resources from only five provinces These provinces were from the southeast region, which led to low genetic diversity. The twenty-three wild centipedegrass accessions in the present study were collected from a wide geographical distribution, accounting for their higher genetic diversity. Accordingly, the high genetic diversity of centipedegrass germplasm may be related to its geographical distribution and biological characteristics. Indeed, centipedegrass is a perennial cross-pollination plant with self-incompatibility (*Hanna & Burton, 1978*). Furthermore, its wide geographical distribution accounts for its adaptability to different ecological environments. In addition, the seed-setting rate of wild centipedegrass germplasm is low. Accordingly, the characteristic of stolon asexual reproduction derived from long-term adaptation and evolution can help maintain the population's genetic diversity (*Hanna, 1995*; *Liu, Hanna & Elsner, 2003*).

The clustering results showed significant differences between the Sichuan population and the non-Sichuan population, and other materials except Sichuan were clustered into one group. The reason for this result may be due to human factors, that is, the species resources in a certain area are brought to another place to grow, and then the gene exchange occurs, which is consistent with the research results of *Elymus nutans* (*Chen et al., 2009*). Our AMOVA results showed a certain degree of genetic differentiation among different geographic populations (Fst = 0.115), which was higher than previous studies (0.0643) (*Susana et al., 2012*), it is also higher than *Yi et al. (2021)* in prairie grass species (0.045). Usually, genetic differentiation is caused by the lack of effective gene exchange. Centipedegrass has a wide geographical distribution, including Jiangsu, Zhejiang, Fujian, Hunan, Hubei and other regions, which limits the gene exchange between distant germplasm and may lead to a certain degree of genetic differentiation between different geographical populations. Our results also substantiated a significant correlation between geographical and genetic distances ($r = 0.385352$, $p = 0.000140$), indicating that geographical isolation led to genetic differentiation, similar to findings reported by *Chen et al. (2020)*. The geographical distribution of the twenty-three accessions collected in this study was relatively dispersed, and the geographical distance was heterogeneous. These factors affected the gene exchange between geographical groups, resulting in greater genetic differences. In addition, climate can lead to genetic variation through natural selection, and environmental adaptability is an important factor in genetic differentiation. Our results showed a significant correlation between BIO14, BIO15, BIO17, and genetic distance; these three climatic factors are associated with rainfall. This finding may be because only some genotypes may survive and prevail with rainfall, which may lead to a decrease in genetic diversity among species populations and even altered gene interactions, consistent with findings reported by *Tan et al. (2018)*. The results showed that there was no significant correlation between seven morphological characteristics and SRAP markers in this study. This is similar to the study of *Zhang et al. (2020)* in *Capsicum annuum*. The reason may be that, first of all, the number of molecular markers obtained in this experiment is small, the number of effective loci is small, and most of the traits are usually regulated by many alleles, so there is a great chance that these loci will not be associated with traits (*He et al., 2023*; *Qu et al., 2023*). Secondly, some loci may be located in some specific and very few regions related to these traits. These loci may have a certain correlation with the region, but the relationship between the two is weak (*Aini et al., 2022*; *Zhang et al., 2021*). We will use some high-throughput sequencing methods such as simplified genome sequencing and resequencing to study in order to obtain better results in future research.

## CONCLUSION

This study evaluated the genetic diversity and population genetic structure of twenty-three wild centipedegrass accessions by SRAP, PCoA, UPGMA and AMOVA. The UPGMA tree map divided all accessions into two clusters, which was roughly consistent with the results of PCoA. AMOVA revealed that the genetic variation within geographical groups was greater than between geographical groups. Overall, the findings of this study can help

better understand the genetic diversity of centipedegrass and lay the groundwork for future research.

### Funding

This research was supported by the Seed Industry Vitalization Research Projects of Jiangsu Province (JBGS[2021]096) and the National Natural Science Foundation of China (32071885). The funders had no role in study design, data collection and analysis, decision to publish, or preparation of the manuscript.

### Grant Disclosures

The following grant information was disclosed by the authors:
Seed Industry Vitalization Research Projects of Jiangsu Province: JBGS[2021]096.
National Natural Science Foundation of China: 32071885.

### Competing Interests

The authors declare there are no competing interests.

### Author Contributions

- Xiaoyun Wang performed the experiments, analyzed the data, prepared figures and/or tables, authored or reviewed drafts of the article, and approved the final draft.
- Wenlong Gou performed the experiments, analyzed the data, authored or reviewed drafts of the article, contributed to the sampling, and approved the final draft.
- Ting Wang performed the experiments, analyzed the data, prepared figures and/or tables, and approved the final draft.
- Yanli Xiong analyzed the data, authored or reviewed drafts of the article, and approved the final draft.
- Yi Xiong performed the experiments, prepared figures and/or tables, and approved the final draft.
- Qingqing Yu performed the experiments, prepared figures and/or tables, and approved the final draft.
- Zhixiao Dong analyzed the data, prepared figures and/or tables, and approved the final draft.
- Xiao Ma conceived and designed the experiments, authored or reviewed drafts of the article, and approved the final draft.
- Nanqing Liu conceived and designed the experiments, authored or reviewed drafts of the article, and approved the final draft.
- Junming Zhao conceived and designed the experiments, analyzed the data, authored or reviewed drafts of the article, and approved the final draft.

### Data Availability

The raw measurements are available in the Supplementary Files.
## Supplemental Information

Supplemental information for this article can be found online at http://dx.doi.org/10.7717/peerj.15900#supplemental-information.

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
