# Peer review of "Genetic diversity analysis and molecular characteristics of wild centipedegrass using sequence-related amplified polymorphism (SRAP) markers"

_PeerJ, doi:10.7717/peerj.15900_

## Round 0.1 · original submission · Minor Revisions

Dear Authors
The manuscript cannot be accepted for publication in its current form. It needs a minor revision to be reconsidered for publication. The authors are invited to revise the paper considering all the suggestions made by the reviewers. Please note that requested changes are required for publication.

Additional comments:
- Authors should scan the manuscript for minor punctuation and English errors (see attached).
- It is highly recommended to correlate the Seven morphological traits measured in 23 centipedegrass and the SRAP data.
- Change The word phenotype in the keywords; it does not refer to anything.
- Provide more details about the methods used to measure the morphological traits.
- As SRAP mentioned for the first time in the abstract, it must be written without abbreviation.

With Thanks

·

Basic reporting

The manuscript is well written but still needs some minor changes

Experimental design

No comments

Validity of the findings

No comments

Additional comments

I reviewed the paper titled "Analyzing genetic diversity and molecular characteristics of wild centipedegrass using sequence-related amplified polymorphism (SRAP) markers". The authors aimed to reveal the population genetic structure of of centipedegrass at the molecular level. Besides, assessment of morphological diversity to obtain more comprehensive information, which is of great significance for preserving valuable genetic resources, selecting high-quality germplasm resources and developing new varieties.
-Comments and Suggestions for Authors
Title
I suggest authors to make the title "Genetic diversity analysis and molecular characterization of wild centipedegrass using sequence-related amplified polymorphism (SRAP) markers"
Abstract
- Please find more corrections as track changes in the manuscript pdf file.
Introduction
-The introduction section is comprehensive and well written.
-Please find more corrections as track changes in the manuscript pdf file.
Materials and methods
-Please find more corrections as track changes in the manuscript pdf file.
Results
-The results section is well written.
- Please find more corrections as track changes in the manuscript pdf file.
Discussion
-The discussion section is well written but I suggest the authors to remove the subtitles from the discussion section.
Conclusion
-The conclusion section is well written.
References
Please unify the style according to the journal instructions

Reviewer 2 ·

Basic reporting

The manuscript entitled "Analyzing genetic diversity and molecular characteristics of wild centipedegrass using sequence-related amplified polymorphism (SRAP) markers"
in the scope of the journal, it was written in good scientific editing and I recommend publishing after minor revision:
1. Introduction: need more explanation and importance of the studied plant with recent references.
2. Results: Authors can add table illustrate different pca for the molecular study.
Discussion need more information, compare your result with previous related species.

Experimental design

Materials showing the collection of plant material, if you can add map for the collection study area

Validity of the findings

Result is good and you can add table showing different pca values

Additional comments

References need to be in 2023

·

Basic reporting

With a little revision, the article will be ready

Experimental design

Table 1 analysis of variance (molecular variance is not correct)

Validity of the findings

Important findings

Additional comments

The article is important

---

## Round 0.2 · accepted · Accept

Dear Authors

I am pleased to inform you that after the last round of revision, the manuscript has been improved a lot, and it can be accepted for publication.

Congratulations on the acceptance of your manuscript, and thank you for your
interest in submitting your work to PeerJ.

The Section Editor noted some specific corrections:

LINE NO: / BEFORE / AFTER / [COMMENTS]
LINE 173: / The present / In the present / [.]
LINE 174: / Fifty-five / fifty-five / [.]
LINE 175: / Fifty-five / fifty-five / [.]

·

Basic reporting

The manuscript is well revised

Experimental design

No comments

Validity of the findings

No comments

Additional comments

I reviewed the paper titled "Analyzing genetic diversity and molecular characteristics of wild centipedegrass using sequence-related amplified polymorphism (SRAP) markers". The authors had improved the manuscript, I recommend that this manuscript should be accepted as is.

Reviewer 2 ·

Basic reporting

The paper is good

Experimental design

In Scientific design

Validity of the findings

Good

Additional comments

No

·

Basic reporting

no comment

Experimental design

no comment

Validity of the findings

no comment

Additional comments

The article is important and accepted